# Demixing odors — fast inference in olfaction

**Agnieszka Grabska-Barwińska**
Gatsby Computational Neuroscience Unit
UCL
agnieszka@gatsby.ucl.ac.uk

**Jeff Beck**
Duke University
jeff@gatsby.ucl.ac.uk

**Alexandre Pouget**
University of Geneva
Alexandre.Pouget@unige.ch

**Peter E. Latham**
Gatsby Computational Neuroscience Unit
UCL
pel@gatsby.ucl.ac.uk

## Abstract

The olfactory system faces a difficult inference problem: it has to determine what odors are present based on the distributed activation of its receptor neurons. Here we derive neural implementations of two approximate inference algorithms that could be used by the brain. One is a variational algorithm (which builds on the work of Beck. et al., 2012), the other is based on sampling. Importantly, we use a more realistic prior distribution over odors than has been used in the past: we use a "spike and slab" prior, for which most odors have zero concentration. After mapping the two algorithms onto neural dynamics, we find that both can infer correct odors in less than 100 ms. Thus, at the behavioral level, the two algorithms make very similar predictions. However, they make different assumptions about connectivity and neural computations, and make different predictions about neural activity. Thus, they should be distinguishable experimentally. If so, that would provide insight into the mechanisms employed by the olfactory system, and, because the two algorithms use very different coding strategies, that would also provide insight into how networks represent probabilities.

## 1  Introduction

The problem faced by the sensory system is to infer the underlying causes of a set of input spike trains. For the olfactory system, the input spikes come from a few hundred different types of olfactory receptor neurons, and the problem is to infer which odors caused them. As there are more than 10,000 possible odors, and more than one can be present at a time, the search space for mixtures of odors is combinatorially large. Nevertheless, olfactory processing is fast: organisms can typically determine what odors are present in a few hundred ms.

Here we ask how organisms could do this. Since our focus is on inference, not learning: we assume that the olfactory system has learned both the statistics of odors in the world and the mapping from those odors to olfactory receptor neuron activity. We then choose a particular model for both, and compute, via Bayes rule, the full posterior distribution. This distribution is, however, highly complex: it tells us, for example, the probability of coffee at a concentration of 14 parts per million (ppm), and no bacon, and a rose at 27 ppm, and acetone at 3 ppm, and no apples and so on, where the "so on" is a list of thousands more odors. It is unlikely that such detailed information is useful to an organism. It is far more likely that organisms are interested in marginal probabilities, such as whether or not coffee is present independent of all the other odors. Unfortunately, even though we can write down the full posterior, calculation of marginal probabilities is intractable due to the

sum over all possible combinations of odors: the number of terms in the sum is exponential in the number of odors.

We must, therefore, consider approximate algorithms. Here we consider two: a variational approximation, which naturally generates approximate posterior marginals, and sampling from the posterior, which directly gives us the marginals. Our main goal is to determine which, if either, is capable of performing inference on ecologically relevant timescales using biologically plausible circuits. We begin by introducing a generative model for spikes in a population of olfactory receptor neurons. We then describe the variational and sampling inference schemes. Both descriptions lead very naturally to network equations. We simulate those equations, and find that both the variational and sampling approaches work well, and require less than 100 ms to converge to a reasonable solution. Therefore, from the point of view of speed and accuracy – things that can be measured from behavioral experiments – it is not possible to rule out either of them. However, they do make different predictions about activity, and so it should be possible to tell them apart from electrophysiological experiments. They also make different predictions about the neural representation of probability distributions. If one or the other could be corroborated experimentally, that would provide valuable insight into how the brain (or at least one part of the brain) codes for probabilities [1].

## 2   The generative model for olfaction

The generative model consists of a probabilistic mapping from odors (which for us are a high level percepts, such as coffee or bacon, each of which consists of a mixture of many different chemicals) to odorant receptor neurons, and a prior over the presence or absence of odors and their concentrations. It is known that each odor, by itself, activates a different subset of the olfactory receptor neurons; typically on the order of 10%-30% [2]. Here we assume, for simplicity, that activation is linear, for which the activity of odorant receptor neuron $i$, denoted $r_i$ is linearly related to the concentrations, $c_j$ of the various odors which are present in a given olfactory scene, plus some background rate, $r_0$. Assuming Poisson noise, the response distribution has the form

$$P(\mathbf{r}|\mathbf{c}) = \prod_i \frac{\left(r_0 + \sum_j w_{ij}c_j\right)^{r_i}}{r_i!} e^{-\left(r_0 + \sum_j w_{ij}c_j\right)}. \tag{2.1}$$

In a nutshell, $r_i$ is Poisson with mean $r_0 + \sum_j w_{ij}c_j$.

In contrast to previous work [3], which used a smooth prior on the concentrations, here we use a spike and slab prior. With this prior, there is a finite probability that the concentration of any particular odor is zero. This prior is much more realistic than a smooth one, as it allows only a small number of odors (out of ~10,000) to be present in any given olfactory scene. It is modeled by introducing a binary variable, $s_j$, which is 1 if odor $j$ is present and 0 otherwise. For simplicity we assume that odors are independent and statistically homogeneous,

$$P(\mathbf{c}|\mathbf{s}) = \prod_j (1 - s_j)\delta(c_j) + s_j\Gamma(c_j|\alpha_1, \beta_1) \tag{2.2a}$$

$$P(\mathbf{s}) = \prod_j \pi^{s_j}(1 - \pi)^{1-s_j} \tag{2.2b}$$

where $\delta(c)$ is the Dirac delta function and $\Gamma(c|\alpha, \beta)$ is the Gamma distribution: $\Gamma(c|\alpha, \beta) = \beta^\alpha c^{\alpha-1}e^{-\beta c}/\Gamma(\alpha)$ with $\Gamma(\alpha)$ the ordinary Gamma function, $\Gamma(\alpha) = \int_0^\infty dx\, x^{\alpha-1}e^{-x}$.

## 3   Inference

### 3.1   Variational inference

Because of the delta-function in the prior, performing efficient variational inference in our model is difficult. Therefore, we smooth the delta-function, and replace it with a Gamma distribution. With this manipulation, the approximate (with respect to the true model, Eq. (2.2a)) prior on $\mathbf{c}$ is

$$P_{var}(\mathbf{c}|\mathbf{s}) = \prod_j (1 - s_j)\Gamma(c_j|\alpha_0, \beta_0) + s_j\Gamma(c_j|\alpha_1, \beta_1). \tag{3.1}$$

The approximate prior allows absent odors to have nonzero concentration. We can partially compensate for that by setting the background firing rate, $r_0$ to zero, and choosing $\alpha_0$ and $\beta_0$ such that the effective background firing rate (due to the small concentration when $s_j = 0$) is equal to $r_0$; see Sec. 4.

As is typical in variational inference, we use a factorized approximate distribution. This distribution, denoted $Q(\mathbf{c}, \mathbf{s}|\mathbf{r})$,was set to $Q(\mathbf{c}|\mathbf{s}, \mathbf{r})Q(\mathbf{s}|\mathbf{r})$ where

$$Q(\mathbf{c}|\mathbf{s}, \mathbf{r}) = \prod_j (1 - s_j)\Gamma(c_j|\alpha_{0j}, \beta_{0j}) + s_j\Gamma(c_j|\alpha_{1j}, \beta_{1j}) \tag{3.2a}$$

$$Q(\mathbf{s}|\mathbf{r}) = \prod_j \lambda_j^{s_j}(1 - \lambda_j)^{1-s_j} . \tag{3.2b}$$

Introducing auxiliary variables, as described in Supplementary Material, and minimizing the Kullback-Leibler distance between $Q$ and the true posterior augmented by the auxiliary variables leads to a set of nonlinear equations for the parameters of $Q$. To simplify those equations, we set $\alpha_1$ to $\alpha_0 + 1$, resulting in

$$\alpha_{0j} = \alpha_0 + \sum_i \frac{r_i w_{ij} F_j(\lambda_j, \alpha_{0j})}{\sum_{k=1} w_{ik} F_k(\lambda_k, \alpha_{0k})} \tag{3.3a}$$

$$L_j \equiv \log \frac{\lambda_j}{1 - \lambda_j} = L_{0j} + \log(\alpha_{0j}/\alpha_0) + \alpha_{0j} \log(\beta_{0j}/\beta_{1j}) \tag{3.3b}$$

where

$$L_{0j} \equiv \log \frac{\pi}{1 - \pi} - \alpha_0 \log(\beta_0/\beta_1) + \log(\beta_1/\beta_{1j}) \tag{3.3c}$$

$$F_j(\lambda, \alpha) \equiv \exp\left[(1 - \lambda)(\Psi(\alpha) - \log \beta_{0j}) + \lambda(\Psi(\alpha + 1) - \log \beta_{1j})\right] \tag{3.3d}$$

and $\Psi(\alpha) \equiv d \log \Gamma(\alpha)/d\alpha$ is the digamma function. The remaining two parameters, $\beta_{0j}$ and $\beta_{1j}$, are fixed by our choice of weights and priors: $\beta_{0j} = \beta_0 + \sum_i w_{ij}$ and $\beta_{1j} = \beta_1 + \sum_i w_{ij}$.

To solve Eqs. (3.3a-b) in a way that mimics the kinds of operations that could be performed by neuronal circuitry, we write down a set of differential equations that have fixed points satisfying Eq. (3.3),

$$\tau_\rho \frac{d\rho_i}{dt} = r_i - \rho_i \sum_j w_{ij} F_j(\lambda_j, \alpha_{0j}) \tag{3.4a}$$

$$\tau_\alpha \frac{d\alpha_{0j}}{dt} = \alpha_0 + F_j(\lambda_j, \alpha_{0j}) \sum_i \rho_i w_{ij} - \alpha_{0j} \tag{3.4b}$$

$$\tau_\lambda \frac{dL_j}{dt} = L_{0j} + \log(\alpha_{0j}/\alpha_0) + \alpha_{0j} \log(\beta_{0j}/\beta_{1j}) - L_j \tag{3.4c}$$

Note that we have introduced an additional variable, $\rho_i$. This variable is proportional to $r_i$, but modulated by divisive inhibition: the fixed point of Eq. (3.4a) is

$$\rho_i = \frac{r_i}{\sum_k w_{ik} F_k(\lambda_k, \alpha_{0k})} . \tag{3.5}$$

Close scrutiny of Eqs. (3.4) and (3.3d) might raise some concerns: (i) $\rho$ and $\alpha$ are reciprocally and symmetrically connected; (ii) there are multiplicative interactions between $F(\lambda_j, \alpha_{0j})$ and $\rho$; and (iii) the neurons need to compute nontrivial nonlinearities, such as logarithm, exponent and a mixture of digamma functions. However: (i) reciprocal and symmetric connectivity exists in the early olfactory processing system [4, 5, 6]; (ii) although multiplicative interactions are in general not easy for neurons, the divisive normalization (Eq. (3.5)) has been observed in the olfactory bulb [7], and (iii) the nonlinearities in our algorithms are not extreme (the logarithm is defined only on the positive range ($\alpha_{0j} > \alpha_0$, Eq. (3.3a)), and $F_j(\lambda, \alpha)$ function is a soft-thresholded linear function; see Fig. S1). Nevertheless, a realistic model would have to approximate Eqs. (3.4a-c), and thus degrade slightly the quality of the inference.

## 3.2 Sampling

The second approximate algorithm we consider is sampling. To sample efficiently from our model, we introduce a new set of variables, $\tilde{c}_j$,

$$c_j = \tilde{c}_j s_j \,. \tag{3.6}$$

When written in terms of $\tilde{c}_j$ rather than $c_j$, the likelihood becomes

$$P(\mathbf{r}|\tilde{\mathbf{c}}, \mathbf{s}) = \prod_i \frac{(r_0 + \sum_j w_{ij}\tilde{c}_j s_j)^{r_i}}{r_i!} e^{-\left(r_0 + \sum_j w_{ij}\tilde{c}_j s_j\right)} \,. \tag{3.7}$$

Because the value of $\tilde{c}_j$ is unconstrained when $s_j = 0$, we have complete freedom in choosing $P(\tilde{c}_j|s_j = 0)$, the piece of the prior corresponding to the absence of odor $j$. It is convenient to set it to the same prior we use when $s_j = 1$, which is $\Gamma(\tilde{c}_j|\alpha_1, \beta_1)$. With this choice, $\tilde{\mathbf{c}}$ is independent of $\mathbf{s}$, and the prior over $\tilde{\mathbf{c}}$ is simply

$$P(\tilde{\mathbf{c}}) = \prod_j \Gamma(\tilde{c}_j|\alpha_1, \beta_1) \,. \tag{3.8}$$

The prior over $\mathbf{s}$, Eq. (2.2b), remains the same. Note that this set of manipulations does not change the model: the likelihood doesn't change, since by definition $\tilde{c}_j s_j = c_j$; when $s_j = 1$, $\tilde{c}_j$ is drawn from the correct prior; and when $s_j = 0$, $\tilde{c}_j$ does not appear in the likelihood.

To sample from this distribution we use Langevin sampling on $\mathbf{c}$ and Gibbs sampling on $\mathbf{s}$. The former is standard,

$$\tau_c \frac{d\tilde{c}_j}{dt} = \frac{\partial \log P(\tilde{\mathbf{c}}, \mathbf{s}|\mathbf{r})}{\partial \tilde{c}_j} + \xi(t) = \frac{\alpha_1 - 1}{\tilde{c}_j} - \beta_1 + s_j \sum_i w_{ij} \left( \frac{r_i}{r_0 + \sum_k w_{ik}\tilde{c}_k s_k} - 1 \right) + \xi(t) \tag{3.9}$$

where $\xi(t)$ is delta-correlated white noise with variance $2\tau$: $\langle \xi_j(t)\xi_{j'}(t') \rangle = 2\tau\delta(t - t')\delta_{jj'}$.

Because the ultimate goal is to implement this algorithm in networks of neurons, we need a Gibbs sampler that runs asynchronously and in real time. This can be done by discretizing time into steps of length $dt$, and computing the update probability for each odor on each time step. This is a valid Gibbs sampler only in the limit $dt \to 0$, where no more than one odor can be updated per time step — that's the limit of interest here. The update rule is

$$T(s_j'|\tilde{\mathbf{c}}, \mathbf{s}, \mathbf{r}) = \nu_0 dt P(s_j'|\tilde{\mathbf{c}}, \mathbf{s}, \mathbf{r}) + (1 - \nu_0 dt)\, \Delta(s_j' - s_j) \tag{3.10}$$

where $s_j' \equiv s_j(t + dt)$, $\mathbf{s}$ and $\tilde{\mathbf{c}}$ should be evaluated at time $t$, and $\Delta(s)$ is the Kronecker delta: $\Delta(s) = 1$ if $s = 0$ and 0 otherwise. As is straightforward to show, this update rule has the correct equilibrium distribution in the limit $dt \to 0$ (see Supplementary Material).

Computing $P(s_j' = 1|\tilde{\mathbf{c}}, \mathbf{s}, \mathbf{r})$ is straightforward, and we find that

$$P(s_j' = 1|\tilde{\mathbf{c}}, \mathbf{s}, \mathbf{r}) = \frac{1}{1 + \exp[-\Phi_j]}$$

$$\Phi_j = \log \frac{\pi}{1 - \pi} + \sum_i \left[ r_i \log \frac{r_0 + \sum_{k \neq j} w_{ik}\tilde{c}_k s_k + w_{ij}\tilde{c}_j}{r_0 + \sum_{k \neq j} w_{ik}\tilde{c}_k s_k} - \tilde{c}_j w_{ij} \right] \,. \tag{3.11}$$

Equations (3.9) and (3.11) raise almost exactly the same concerns that we saw for the variational approach: (i) $c$ and $s$ are reciprocally and symmetrically connected; (ii) there are multiplicative interactions between $\tilde{c}$ and $s$; and (iii) the neurons need to compute nontrivial nonlinearities, such as logarithm and divisive normalization. Additionally, the noise in the Langevin sampler ($\xi$ in Eq. 3.9) has to be white and have exactly the right variance. Thus, as with the variational approach, we expect a biophysical model to introduce approximations, and, therefore — as with the variational algorithm — degrade slightly the quality of the inference.

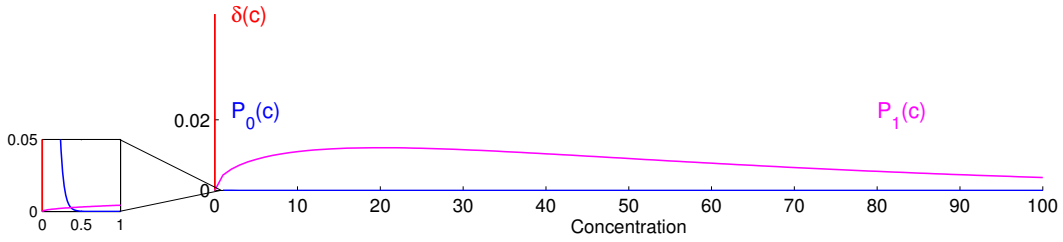

Figure 1: Priors over concentration. The true priors – the ones used to generate the data – are shown in red and magenta; these correspond to $\delta(c)$ and $\Gamma(c|\alpha_1, \beta_1)$, respectively. The variational prior in the absence of an odor, $\Gamma(c|\alpha_0, \beta_0)$ with $\alpha_0 = 0.5$ and $\beta_0 = 20$, is shown in blue.

## 4 Simulations

To determine how fast and accurate these two algorithms are, we performed a set of simulations using either Eq. (3.4) (variational inference) or Eqs. (3.9 - 3.11) (sampling). For both algorithms, the odors were generated from the true prior, Eq. (2.2). We modeled a small olfactory system, with 40 olfactory receptor types (compared to approximately 350 in humans and 1000 in mice [8]). To keep the ratio of identifiable odors to receptor types similar to the one in humans [8], we assumed 400 possible odors, with 3 odors expected to be present in the scene ($\pi = 3/400$). If an odor was present, its concentration was drawn from a Gamma distribution with $\alpha_1 = 1.5$ and $\beta_1 = 1/40$. The background spike count, $r_0$, was set to 1. The connectivity matrix was binary and random, with a connection probability, $p_c$ (the probability that any particular element is 1), set to 0.1 [2]. All network time constants ($\tau_\rho, \tau_\alpha, \tau_\lambda, \tau_c$ and $1/\nu_0$, from Eqs (3.4), (3.9) and (3.10)) were set to 10 ms. The differential equations were solved using the Euler method with a time step of 0.01 ms. Because we used $\alpha_1 = \alpha_0 + 1$, the choice $\alpha_1 = 1.5$ forced $\alpha_0$ to be 0.5. Our remaining parameter, $\beta_0$, was set to ensure that, for the variational algorithm, the absent odors (those with $s_j = 0$) contributed a background firing rate of $r_0$ on average. This average background rate is given by $\sum_j \langle w_{ij} \rangle \langle c_j \rangle = p_c N_{odors} \alpha_0/\beta_0$. Setting this to $r_0$ yields $\beta_0 = p_c N_{odors} \alpha_0/r_0 = 0.1 \times 400 \times 0.5/1 = 20$. The true (Eq. (2.2)) and approximate (Eq. (3.1)) prior distributions over concentration are shown in Fig. 1.

Figure 2 shows how the inference process evolves over time for a typical set of odors and concentrations. The top panel shows concentration, with variational inference on the left (where we plot the mean of the posterior distribution over concentration, $(1 - \lambda_j)\alpha_{0j}(t)/\beta_{0j}(t) + \lambda_j \alpha_{1j}(t)/\beta_{1j}(t)$; see Eq. (3.2)) and sampling on the right (where we plot $\tilde{c}_j$, the output of our Langevin sampler; see Eq. (3.9)) for a case with three odors present. The three colored lines correspond to the odors that

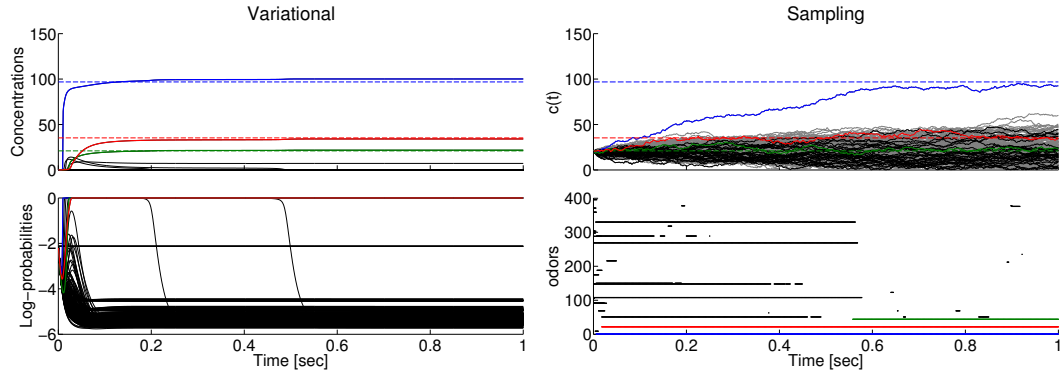

Figure 2: Example run for the variational algorithm (left) and sampling (right); see text for details. In the bottom left panel the green, blue and red lines go to a probability of 1 ( log probability of 0) within about 50 ms. In sampling, the initial value of concentrations is set to the most likely value under the prior ($\tilde{c}(0) = (\alpha_1 - 1)/\beta_1$). The dashed lines are the true concentrations.

were presented, with solid lines for the inferred concentrations and dashed lines for the true ones. Black lines are the odors that were not present. At least in this example, both algorithms converge rapidly to the true concentration.

In the bottom left panel of Fig. 2 we plot the log-probability that each of the odors is present, $\lambda_j(t)$. The present odors quickly approach probabilities of 1; the absent odors all have probabilities below $10^{-4}$ within about 200 ms. The bottom right panel shows samples from $s_j$ for all the odors, with dots denoting present odors ($s_j(t) = 1$) and blanks absent odors ($s_j(t) = 0$). Beyond about 500 ms, the true odors (the colored lines at the bottom) are on continuously, and for the odors that were not present, $s_j$ is still occasionally 1, but relatively rarely.

In Fig. 3 we show the time course of the probability of odors when between 1 and 5 odors were presented. We show only the first 100 ms, to emphasize the initial time course. Again, variational inference is on the left and sampling is on the right. The black lines are the average values of the probability of the correct odors; the gray regions mark 25%–75% percentiles. Ideally, we would like to compare these numbers to those expected from a true posterior. However, due to its intractability, we must seek different means of comparison. Therefore, we plot the probability of the most likely non-presented odor (red); the average probability of the non-presented odors (green), and the probability of guessing the correct odors via simple template matching (dashed; see Fig. 3 legend for details).

Although odors are inferred relatively rapidly (they exceed template matching within 20 ms), there were almost always false positives. Even with just one odor present, both algorithms consistently report the existence of another odor (red). This problem diminishes with time if fewer odors are presented than the expected three, but it persists for more complex mixtures. The false positives are in fact consistent with human behavior: humans have difficulty correctly identify more than one odor in a mixture, with the most common problem being false positives [9].

Finally, because the two algorithms encode probabilities differently (see Discussion below), we also look into the time courses of the neural activity. In Fig. 4, we show the log-probability, $L$ (left), and probability, $\lambda$ (right), averaged across 400 scenes containing 3 odors (see Supplementary Fig. 2 for the other odor mixtures). The probability of absent odors drops from $\log(3/400) \approx e^{-5}$ (the prior) to $e^{-12}$ (the final inferred probability). For the variational approach, this represents a drop in activity of 7 log units, comparable to the increase of about 5 log units for the present odors (whose probability is inferred to be near 1). For the sampling based approach, on the other hand, this represents a very small drop in activity. Thus, for the variational algorithm the average activity associated with the absent odors exhibits a large drop, whereas for the sampling based approach the average activity associated with the absent odors starts small and stays small.

## 5    Discussion

We introduced two algorithms for inferring odors from the activity of the odorant receptor neurons. One was a variational method; the other sampling based. We mapped both algorithms onto dynamical systems, and, assuming time constants of 10 ms (plausible for biophysically realistic networks), tested the time course of the inference.

The two algorithms performed with striking similarity: they both inferred odors within about 100 ms and they both had about the same accuracy. However, since the two methods encode probabilities differently (linear vs logarithmic encoding), they can be differentiated at the level of neural activity. As can be seen by examining Eqs. (3.4a) and (3.4c), for variational inference the log probability of concentration and presence/absence are related to the dynamical variables via

$$\log Q(c_j) \sim \alpha_{1j} \log c_j - \beta_{1j} c_j \tag{5.1a}$$

$$\log Q(s_j) \sim L_j s_j \tag{5.1b}$$

where $\sim$ indicates equality within a constant. If we interpret $\alpha_{0j}$ and $L_j$ as firing rates, then these equations correspond to a linear probabilistic population code [10]: the log probability inferred by the approximate algorithm is linear in firing rate, with a parameter-dependent offset (the term $-\beta_{1j} c_j$ in Eq. (5.1a)). For the sampling-based algorithm, on the other hand, activity generates samples from the posterior; an average of those samples codes for the probability of an odor being present. Thus, if the olfactory system uses variational inference, activity should code for log probability, whereas if it uses sampling, activity should code for probability.

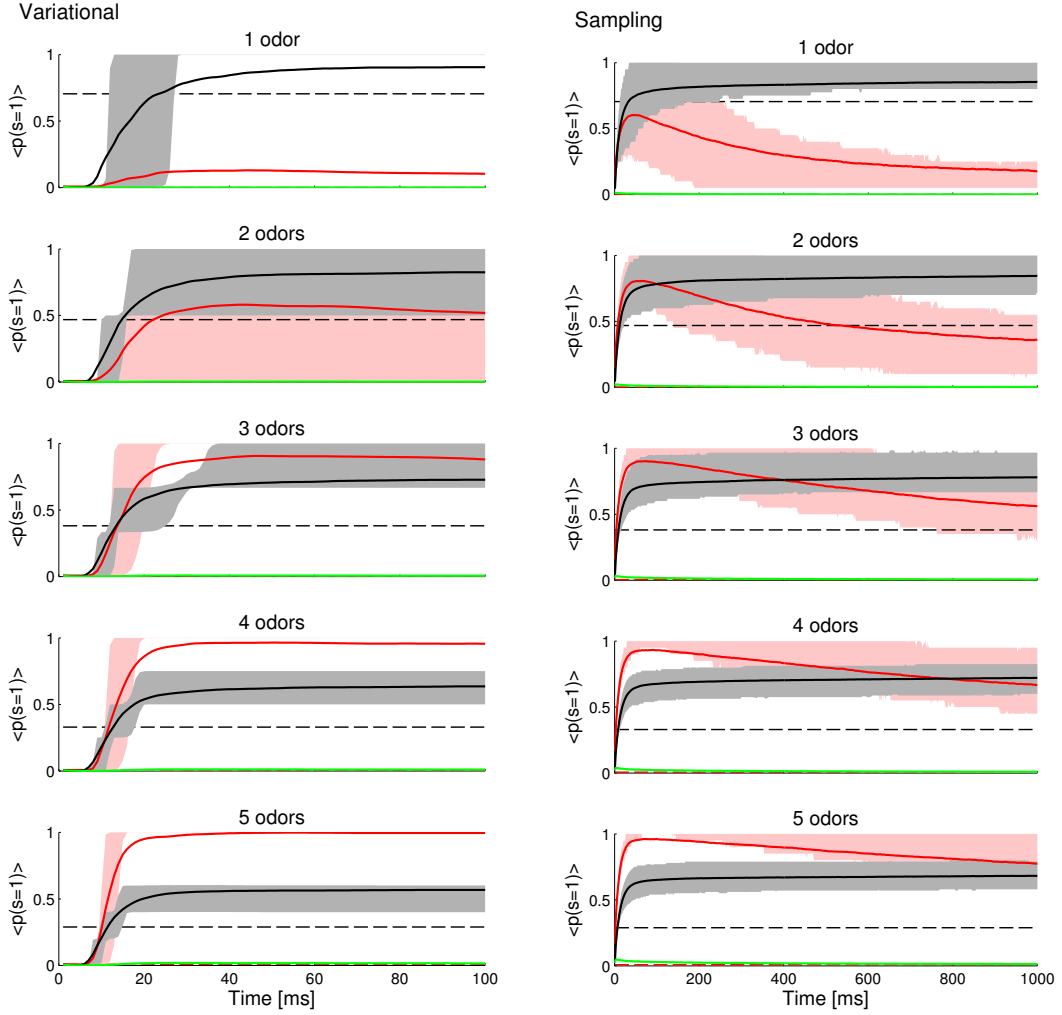

Figure 3: Inference by networks — initial 100 ms. Black: average value of the probability of correct odors; red: probability of the most likely non-presented odor; green: average probability of the non-presented odors. Shaded areas represent 25th–75th percentile of values across 400 olfactory scenes. In the variational approach, values are often either 0 or 1, which makes it possible for the mean to land outside of the chosen percentile range; this happens whenever the odors are guessed correctly more than 75% of the time, in which case the 25th–75th percentile collapses to 1, or less than 25% of the time, in which case the 25th–75th percentile collapses to 0. The left panel shows variational inference, where we plot $\lambda_j(t)$; the right one shows sampling, where we plot $s_k(t)$ averaged over 20 repetitions of the algorithm (to avoid arbitrariness in decoding/smoothing/averaging the samples). Both methods exceed template matching within 20 ms (dashed line). (Template matching finds odors (the $j$'s) that maximize the dot product between the activity, $r_i$, and the weights, $w_{ij}$, associated, with odor $j$; that is, it chooses $j$'s that maximize $\sum_i r_i w_{ij} / \left( \sum_i r_i^2 \sum_i w_{ij}^2 \right)^{1/2}$. The number of odors chosen by template matching was set to the number of odors presented.) For more complex mixtures, sampling is slightly more efficient at inferring the presented odors. See Supplementary Material for the time course out to 1 second and for mixtures of up to 10 odors.

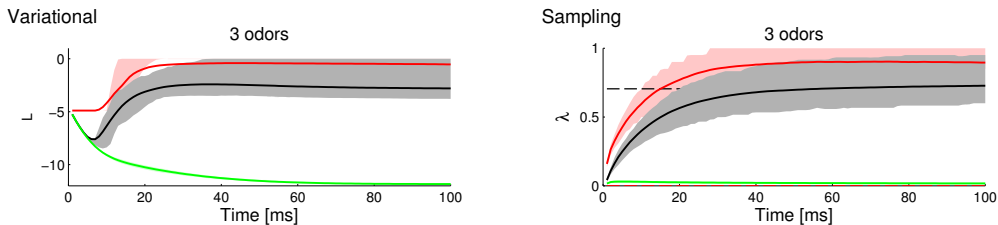

Figure 4: Average time course of $\log(p(s))$ (left) and $p(s)$ (right, same as in Fig. 3). For the variational algorithm, the activity of the neurons codes for log probability (relative to some background to keep firing rates non-negative). For this algorithm, the drop in probability of the non-presented odors from about $e^{-5}$ to $e^{-12}$ corresponds to a large drop in firing rate. For the sampling based algorithm, on the other hand, activity codes for probability, and there is almost no drop in activity.

There are two ways to determine which. One is to note that for the variational algorithm there is a large drop in the average activity of the neurons coding for the non-present odors (Fig. 4 and Supplementary Figure 2). This drop could be detected with electrophysiology. The other focuses on the present odors, and requires a comparison between the posterior probability inferred by an animal and neural activity. The inferred probability can be measured by so-called "opt-out" experiments [11]; the latter by sticking an electrode into an animal's head, which is by now standard.

The two algorithms also make different predictions about the activity coding for concentration. For the variational approach, activity, $\alpha_{0j}$, codes for the parameters of a probability distribution. Importantly, in the variational scheme the mean and variance of the distribution are tied – both are proportional to activity. Sampling, on the other hand, can represent arbitrary concentration distributions. These two schemes could, therefore, be distinguished by separately manipulating average concentration and uncertainty – by, for example, showing either very similar or very different odors.

Unfortunately, it is not clear where exactly one needs to stick the electrode to record the trace of the olfactory inference. A good place to start would be the olfactory bulb, where odor representations have been studied extensively [12, 13, 14]. For example, the dendro-dendritic connections observed in this structure [4] are particularly well suited to meet the symmetry requirements on $w_{ij}$. We note in passing that these connections have been the subject of many theoretical studies. Most, however, considered single odors [15, 6, 16], for which one does not need a complicated inference process An early notable exception to the two-odor standard was Zhaoping [17], who proposed a model for serial analysis of complex mixtures, whereby higher cortical structures would actively adapt the already recognized components and send a feedback signal to the lower structures. Exactly how her network relates to our inference algorithms remains unclear. We should also point out that although the olfactory bulb is a likely location for at least part of our two inference algorithms, both are sufficiently complicated that they may need to be performed by higher cortical structures, such as the anterior piriform cortex, [18, 19].

**Future directions.** We have made several unrealistic assumptions in this analysis. For instance, the generative model was very simple: we assumed that concentrations added linearly, that weights were binary (so that each odor activated a subset of the olfactory receptor neurons at a finite value, and did not activate the rest at all), and that noise was Poisson. None of these are likely to be exactly true. And we considered priors such that all odors were independent. This too is unlikely to be true – for instance, the set of odors one expects in a restaurant are very different than the ones one expects in a toxic waste dump, consistent with the fact that responses in the olfactory bulb are modulated by task-relevant behavior [20]. Taking these effects into account will require a more complicated, almost certainly hierarchical, model. We have also focused solely on inference: we assumed that the network knew perfectly both the mapping from odors to odorant receptor neurons and the priors. In fact, both have to be learned. Finally, the neurons in our network had to implement relatively complicated nonlinearities: logs, exponents, and digamma and quadratic functions, and neurons had to be reciprocally connected. Building a network that can both exhibit the proper nonlinearities (at least approximately) and learn the reciprocal weights is challenging. While these issues are nontrivial, they do not appear to be insurmountable. We expect, therefore, that a more realistic model will retain many of the features of the simple model we presented here.

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
