[Supplementary Material]

# Demixing smells — fast inference in olfaction Supplementary Material

**Agnieszka Grabska-Barwińska**
Gatsby Computational Neuroscience Unit
UCL
agnieszka@gatsby.ucl.ac.uk

**Jeff Beck**
Duke University
jeff@gatsby.ucl.ac.uk

**Alexandre Pouget**
University of Geneva
Alexandre.Pouget@unige.ch

**Peter E. Latham**
Gatsby Computational Neuroscience Unit
UCL
pel@gatsby.ucl.ac.uk

## 1 Variational Bayesian inference

Here we derive the equations we used in the main text for variational inference. There are three parts to the derivation: first we modify slightly our original generative model, then we introduce a set of auxiliary variables (or, more accurately, they introduce themselves), and, finally, we write down a variational approximation that takes into account the auxiliary variables.

### 1.1 The variational generative model

We start, mainly for completeness, by writing down the original generative model; this is the same as Eqs. (2.1) and (2.2) of the main text. For the likelihood we assume that the spike counts are Poisson,

$$P(\mathbf{r}|\mathbf{c}) = \prod_i \frac{\left(r_0 + \sum_j w_{ij}c_j\right)^{r_i}}{r_i!} e^{-\left(r_0 + \sum_j w_{ij}c_j\right)} \tag{1.1a}$$

and we use a "spike and slab" prior on the concentrations and a Bernoulli prior on $s_j$, the variable that indicates presence or absence of odor $j$,

$$P(\mathbf{c}|\mathbf{s}) = \prod_j (1 - s_j)\delta(c_j) + s_j\Gamma(c_j|\alpha_1, \beta_1) \tag{1.1b}$$

$$P(\mathbf{s}) = \prod_j \pi^{s_j}(1 - \pi)^{1 - s_j} \tag{1.1c}$$

where $\delta(c)$ is the Dirac delta-function and $\Gamma(c|\alpha, \beta)$ is the Gamma distribution,

$$\Gamma(c|\alpha, \beta) = \frac{\beta^\alpha c^{\alpha-1} e^{-\beta c}}{\Gamma(\alpha)} . \tag{1.2}$$

Here $\Gamma(\alpha)$ is the ordinary Gamma function: $\Gamma(\alpha) = \int_0^\infty dx \, x^{\alpha-1} e^{-x}$.

Because of the delta-function in the prior, performing efficient variational inference in our model is, as far as we know, difficult. Therefore, we smooth the delta-function, and replace it with a Gamma distribution, $\delta(c_j) \longrightarrow \Gamma(c_j|\alpha_0, \beta_0)$. In the limit $\alpha_0 \to 0$ and $\beta_0 \to \infty$ we recover the true prior. However, when these two parameters are finite, absent odors – odors with $s_j = 0$ – have non-zero concentrations. To correct for this, we choose $\alpha_0$ and $\beta_0$ so that there is an effective background

rate equal to $r_0$, and then we remove $r_0$ from the likelihood. This results in a variational generative model of the form

$$P_{var}(\mathbf{r}|\mathbf{c}) = \prod_i \frac{\left(\sum_j w_{ij}c_j\right)^{r_i}}{r_i!} e^{-\left(\sum_j w_{ij}c_j\right)} \tag{1.3a}$$

$$P_{var}(\mathbf{c}|\mathbf{s}) = \prod_j (1 - s_j)\Gamma(c_j|\alpha_0, \beta_0) + s_j\Gamma(c_j|\alpha_1, \beta_1) \tag{1.3b}$$

$$P_{var}(\mathbf{s}) = \prod_j \pi^{s_j}(1 - \pi)^{1-s_j} . \tag{1.3c}$$

To choose $\alpha_0$ and $\beta_0$ to mimic the background firing rate, $r_0$, we note that if all the odors were absent (all the $s_j$ were zero), the average background rate, $r_{average}$, would be

$$r_{average} = \sum_j \langle w_{ij}\rangle\langle c_j\rangle = p_c N_{odors}\alpha_0/\beta_0 \tag{1.4}$$

where $p_c$ is the connection probability and $N_{odors}$ is the total number of odors (recall that $w_{ij}$ is 1 with probability $p_c$ and 0 with probability $1 - p_c$; see main text, Sec. 4), and $\alpha_0/\beta_0$ is the average value of $c_j$ under the prior when $s_j = 0$. Setting $r_{average}$ to $r_0$ implies that

$$\frac{\alpha_0}{\beta_0} = \frac{r_0}{p_c N_{odors}} . \tag{1.5}$$

We enforce this constraint in our simulations.

## 1.2 Turning products of sums into sums of products

Collecting the terms in Eq. (1.3a), we see that the posterior distribution over $\mathbf{c}$ and $\mathbf{s}$ is given by

$$P(\mathbf{s}, \mathbf{c}|\mathbf{r}) \propto \prod_i \frac{\left(\sum_j w_{ij}c_j\right)^{r_i}}{r_i!} e^{-\left(\sum_j w_{ij}c_j\right)} \tag{1.6}$$
$$\times \prod_j \left[(1 - s_j)\Gamma(c_j|\alpha_0, \beta_0) + s_j\Gamma(c_j|\alpha_1, \beta_1)\right]\left[\pi^{s_j}(1 - \pi)^{1-s_j}\right].$$

Variational inference with this posterior is hard, primarily because the likelihood consists of products over sums. We can, however, turn those products over sums into sums over products by using the multinomial theorem,

$$\left(\sum_j w_{ij}c_j\right)^{r_i} = \sum_{N_{ij}} \Delta\left(r_i - \sum_j N_{ij}\right) r_i! \prod_{j=0} \frac{(w_{ij}c_j)^{N_{ij}}}{N_{ij}!} \tag{1.7}$$

where $\Delta$ is the the Kronecker delta, $\Delta(n) = 1$ if $n = 0$ and 0 otherwise, and the sum over $N_{ij}$ is shorthand for a set of sums in which $N_{i1}$, $N_{i2}$, ... all run from 0 to $r_i$. The posterior distribution can now be written

$$P(\mathbf{c}, \mathbf{s}|\mathbf{r}) = \sum_{\mathbf{N}} P(\mathbf{N}, \mathbf{c}, \mathbf{s}|\mathbf{r}) \tag{1.8}$$

where, inserting Eq. (1.3b) into (1.6), $P(\mathbf{N}, \mathbf{c}, \mathbf{s}|\mathbf{r})$ is given by

$$P(\mathbf{N}, \mathbf{c}, \mathbf{s}|\mathbf{r}) \propto \prod_i \Delta\left(r_i - \sum_j N_{ij}\right) \prod_j \frac{(w_{ij}c_j)^{N_{ij}} e^{-w_{ij}c_j}}{N_{ij}!} \tag{1.9}$$
$$\times \prod_j \left[(1 - s_j)\Gamma(c_j|\alpha_0, \beta_0) + s_j\Gamma(c_j|\alpha_1, \beta_1)\right]\left[\pi^{s_j}(1 - \pi)^{1-s_j}\right].$$

## 1.3 The variational approximation

The variational approach we use approximates the augmented posterior distribution, $P(\mathbf{N}, \mathbf{c}, \mathbf{s}|\mathbf{r})$, rather than the original one, $P(\mathbf{c}, \mathbf{s}|\mathbf{r})$. We use a factorized variational distribution of the form

$$Q(\mathbf{N}, \mathbf{c}, \mathbf{s}|\mathbf{r}) = Q(\mathbf{N}|\mathbf{r})Q(\mathbf{c}, \mathbf{s}|\mathbf{r}) \tag{1.10}$$

where we are using the notation that a probability distribution is labeled by its argument. This can in principle produce confusion, but it won't for this problem.

Our goal is to choose $Q(\mathbf{N}, \mathbf{c}, \mathbf{s}|\mathbf{r})$ so that it minimizes the KL distance between $Q(\mathbf{N}, \mathbf{c}, \mathbf{s}|\mathbf{r})$ and $P(\mathbf{N}, \mathbf{c}, \mathbf{s}|\mathbf{r})$. To see what this implies, we explicitly minimize the KL distance with respect to $Q(\mathbf{N}|\mathbf{r})$. To do that we first differentiate with respect to $Q(\mathbf{N}|\mathbf{r})$,

$$\frac{d}{dQ(\mathbf{N}|\mathbf{r})} \sum_{\mathbf{N},\mathbf{s}} \int d\mathbf{c} \, Q(\mathbf{N}, \mathbf{c}, \mathbf{s}|\mathbf{r}) \log \frac{Q(\mathbf{N}, \mathbf{c}, \mathbf{s}|\mathbf{r})}{P(\mathbf{N}, \mathbf{c}, \mathbf{s}|\mathbf{r})} \tag{1.11}$$

$$= 1 + \log Q(\mathbf{N}|\mathbf{r}) - \sum_{\mathbf{s}} \int d\mathbf{c} \, Q(\mathbf{c}, \mathbf{s}|\mathbf{r}) \log P(\mathbf{N}, \mathbf{c}, \mathbf{s}|\mathbf{r}) \,,$$

and then set the right hand side to zero. This yields

$$\log Q(\mathbf{N}|\mathbf{r}) \sim \langle \log P(\mathbf{N}, \mathbf{c}, \mathbf{s}|\mathbf{r}) \rangle_{Q(\mathbf{c},\mathbf{s}|\mathbf{r})} \tag{1.12a}$$

where "$\sim$" indicates equality up to constants. An essentially identical calculation yields

$$\log Q(\mathbf{c}, \mathbf{s}|\mathbf{r}) \sim \langle \log P(\mathbf{N}, \mathbf{c}, \mathbf{s}|\mathbf{r}) \rangle_{Q(\mathbf{N}|\mathbf{r})} \,. \tag{1.12b}$$

To proceed, we simply need to average $\log P(\mathbf{N}, \mathbf{c}, \mathbf{s}|\mathbf{r})$ with respect to the variational distributions. We start by writing down an explicit expression for $\log P(\mathbf{N}, \mathbf{c}, \mathbf{s}|\mathbf{r})$,

$$\log P(\mathbf{N}, \mathbf{c}, \mathbf{s}|\mathbf{r}) \sim \sum_i \log \Delta\left(r_i - \sum_j N_{ij}\right) + \sum_{ij} N_{ij} \log(w_{ij} c_j) - w_{ij} c_j - \log N_{ij}!$$

$$+ \sum_j (1 - s_j)\left[(\alpha_0 - 1)\log c_j - \beta_0 c_j\right] + s_j\left[(\alpha_1 - 1)\log c_j - \beta_1 c_j\right]$$

$$+ \sum_j s_j \log\left[\frac{\pi}{1 - \pi} \frac{\Gamma(\alpha_0)}{\beta_0^{\alpha_0}} \frac{\beta_1^{\alpha_1}}{\Gamma(\alpha_1)}\right] \,. \tag{1.13}$$

Using Eq. (1.12), and performing averages over either $Q(\mathbf{N}|\mathbf{r})$ or $Q(\mathbf{c}, \mathbf{s}|\mathbf{r})$ in Eq. (1.13), whichever is appropriate, we arrive at

$$\log Q(\mathbf{N}|\mathbf{r}) \sim \sum_i \log \Delta\left(r_i - \sum_j N_{ij}\right) + \sum_{ij} N_{ij} \log w_{ij} + N_{ij} \langle \log c_j \rangle_{Q(\mathbf{c},\mathbf{s}|\mathbf{r})} - \log N_{ij}! \tag{1.14a}$$

$$\log Q(\mathbf{c}, \mathbf{s}|\mathbf{r}) \sim \sum_j (1 - s_j)\left[\left(\alpha_0 + \langle N_{ij} \rangle_{Q(\mathbf{N}|\mathbf{r})} - 1\right)\log c_j - \left(\beta_0 + \sum_i w_{ij}\right)c_j\right]$$

$$+ \sum_j s_j\left[\left(\alpha_1 + \langle N_{ij} \rangle_{Q(\mathbf{N}|\mathbf{r})} - 1\right)\log c_j - \left(\beta_1 + \sum_i w_{ij}\right)c_j\right] \tag{1.14b}$$

$$+ \sum_j s_j \log\left[\frac{\pi}{1 - \pi} \frac{\Gamma(\alpha_0)}{\beta_0^{\alpha_0}} \frac{\beta_1^{\alpha_1}}{\Gamma(\alpha_1)}\right] \,.$$

Examining these expressions, we see that $Q(\mathbf{N}|\mathbf{r})$ is multinomial and $Q(\mathbf{c}, \mathbf{s}|\mathbf{r})$ is the sum of Gamma distributions. Using red to indicate the parameters of these distributions, and noting that $Q(\mathbf{c}, \mathbf{s}|\mathbf{r})$

can be decomposed as $Q(\mathbf{c}, \mathbf{s}|\mathbf{r}) = Q(\mathbf{c}|\mathbf{s}, \mathbf{r})Q(\mathbf{s}|\mathbf{r})$, we have

$$Q(\mathbf{N}|\mathbf{r}) = \prod_i \Delta\left(r_i - \sum_j N_{ij}\right) r_i! \prod_j \frac{p_{ij}^{N_{ij}}}{N_{ij}!} \tag{1.15a}$$

$$Q(\mathbf{c}|\mathbf{s}, \mathbf{r}) = \prod_j \left[(1 - s_j)\Gamma(c_j|\alpha_{0j}, \beta_{0j}) + s_j\Gamma(c_j|\alpha_{1j}, \beta_{1j})\right] \tag{1.15b}$$

$$Q(\mathbf{s}|\mathbf{r}) = \prod_j \lambda_j^{s_j}(1 - \lambda_j)^{s_j} \tag{1.15c}$$

where

$$p_{ij} = \frac{w_{ij}e^{\langle \log c_j \rangle_{Q(\mathbf{c},\mathbf{s}|\mathbf{r})}}}{\sum_j w_{ij}e^{\langle \log c_j \rangle_{Q(\mathbf{c},\mathbf{s}|\mathbf{r})}}} \tag{1.16a}$$

$$\alpha_{0j} = \alpha_0 + \langle N_{ij} \rangle_{Q(\mathbf{N}|\mathbf{r})} \tag{1.16b}$$

$$\alpha_{1j} = \alpha_1 + \langle N_{ij} \rangle_{Q(\mathbf{N}|\mathbf{r})} \tag{1.16c}$$

$$\beta_{0j} = \beta_0 + \sum_i w_{ij} \tag{1.16d}$$

$$\beta_{1j} = \beta_1 + \sum_i w_{ij} \tag{1.16e}$$

$$\frac{\lambda_j}{1 - \lambda_j} = \frac{\pi}{1 - \pi} \frac{\beta_{0j}^{\alpha_{0j}}\Gamma(\alpha_0)}{\beta_0^{\alpha_0}\Gamma(\alpha_{0j})} \frac{\beta_1^{\alpha_1}\Gamma(\alpha_{1j})}{\beta_{1j}^{\alpha_{1j}}\Gamma(\alpha_1)}. \tag{1.16f}$$

Equations (1.15b) and (1.15c) correspond to Eq. (3.2) of the main text.

Now all we have to do is compute $\langle N_{ij} \rangle_{Q(\mathbf{N}|\mathbf{r})}$ and $\langle \log c_j \rangle_{Q(\mathbf{c},\mathbf{s}|\mathbf{r})}$. The former is straightforward: using Eq. (1.15a), we see that

$$\langle N_{ij} \rangle_{Q(\mathbf{N}|\mathbf{r})} = r_i p_{ij}. \tag{1.17}$$

Thus, Eqs. (1.16b) and (1.16c) become

$$\alpha_{0j} = \alpha_0 + \sum_i r_i p_{ij} \tag{1.18a}$$

$$\alpha_{1j} = \alpha_1 + \sum_i r_i p_{ij}. \tag{1.18b}$$

The latter quantity, $\langle \log c_j \rangle_{Q(\mathbf{c},\mathbf{s}|\mathbf{r})}$, is slightly more complicated. Note first of all that

$$\langle \log c_j \rangle_{Q(\mathbf{c},\mathbf{s}|\mathbf{r})} = \sum_{\mathbf{s}} Q(\mathbf{s}|\mathbf{r}) \int d\mathbf{c}\, Q(\mathbf{c}, \mathbf{s}|\mathbf{r}) \log c_j. \tag{1.19}$$

Examining Eqs. (1.15b) and (1.15c), we see that the integral over $\mathbf{c}$ is an integral over Gamma functions. These are known integrals,

$$\langle \log c \rangle_{\Gamma(c|\alpha,\beta)} = \Psi(\alpha) - \log\beta \tag{1.20}$$

where $\Psi$ is the digamma function: $\Psi(\alpha) = d\log\Gamma(\alpha)/d\alpha$. We thus have

$$\langle \log c_j \rangle_{Q(\mathbf{c},\mathbf{s}|\mathbf{r})} = (1 - \lambda_j)(\Psi(\alpha_{0j}) - \log\beta_{0j}) + \lambda_j(\Psi(\alpha_{1j}) - \log\beta_{1j}), \tag{1.21}$$

and $p_{ij}$ becomes

$$p_{ij} = \frac{w_{ij}e^{(1-\lambda_j)(\Psi(\alpha_{0j}) - \log\beta_{0j}) + \lambda_j(\Psi(\alpha_{1j}) - \log\beta_{1j})}}{\sum_k w_{ik}e^{(1-\lambda_j)(\Psi(\alpha_{0k}) - \log\beta_{0k}) + \lambda_k(\Psi(\alpha_{1k}) - \log\beta_{1k})}}. \tag{1.22}$$

When $\alpha_1 = \alpha_0 + 1$, the set of equations for the parameters of the variational distribution simplify considerably, and reduce to Eq. (3.3) of the main text.

Figure 1: $F(\alpha, \lambda)$ versus $\alpha$ for different values of $\lambda$.

In this limit, $\alpha_{1j} = \alpha_{0j} + 1$, and we can write

$$p_{ij} = \frac{w_{ij} F_j(\lambda_j, \alpha_{0j})}{\sum_k w_{ik} F_k(\lambda_k, \alpha_k)} \tag{1.23}$$

with

$$F_j(\lambda, \alpha) \equiv \exp\left[(1 - \lambda)(\Psi(\alpha) - \log \beta_{0j}) + \lambda(\Psi(\alpha + 1) - \log \beta_{1j})\right]. \tag{1.24}$$

This nonlinearity can be further decomposed into

$$F_j(\lambda, \alpha) = (\beta_{1j}/\beta_{0j})^{-\lambda} \beta_{0j}^{-1} \times F(\lambda, \alpha). \tag{1.25}$$

In Figure 1 we plot $F(\lambda, \alpha)$

$$F(\lambda, \alpha) = \exp\left[(1 - \lambda)(\Psi(\alpha)) + \lambda(\Psi(\alpha + 1))\right]. \tag{1.26}$$

This function is essentially threshold linear.

## 2   Sampling

Here we show that Gibbs sampling on the $s_j$ does indeed have the correct equilibrium distribution in the limit $dt \to 0$. We start with the update rule, which comes from Eq. (3.10) of the main text,

$$T(s_j'|\tilde{\mathbf{c}}, \mathbf{s}, \mathbf{r}) = \nu_0 dt P(s_j'|\tilde{\mathbf{c}}, \mathbf{s}, \mathbf{r}) + (1 - \nu_0 dt)\, \Delta(s_j' - s_j) \tag{2.1}$$

where $s_j' \equiv s_j(t + dt)$, and $\mathbf{s}$ and $\tilde{\mathbf{c}}$ should be evaluated at time $t$.

We want to show that this update rule acting on the true distribution maps to the true distribution in the small $dt$ limit. Keeping only terms that are first order in $dt$, we have

$$\sum_{\mathbf{s}} \left[\prod_j T(s_j'|\tilde{\mathbf{c}}, \mathbf{s}, \mathbf{r})\right] P(\tilde{\mathbf{c}}, \mathbf{s}|\mathbf{r}) = (1 - N_{odors}\nu_0 dt) \sum_{\mathbf{s}} \left[\prod_j \Delta(s_j' - s_j)\right] P(\tilde{\mathbf{c}}, \mathbf{s}|\mathbf{r}) \tag{2.2}$$

$$+ \nu_0 dt \sum_{\mathbf{s}} \sum_j \left[\prod_{j' \neq j} \Delta(s_{j'}' - s_{j'})\right] P(s_j'|\tilde{\mathbf{c}}, \mathbf{s}, \mathbf{r}) P(\tilde{\mathbf{c}}, \mathbf{s}|\mathbf{r}).$$

Sums involving $\Delta(s'_j - s_j)$ are trivial, giving us

$$\sum_{\mathbf{s}} \left[ \prod_j T(s'_j|\tilde{\mathbf{c}}, \mathbf{s}, \mathbf{r}) \right] P(\tilde{\mathbf{c}}, \mathbf{s}|\mathbf{r}) = (1 - N_{odors}\nu_0 dt) P(\tilde{\mathbf{c}}, \mathbf{s}'|\mathbf{r}) \tag{2.3}$$

$$+ \nu_0 dt \sum_j \sum_{s_j} P(s'_j|\tilde{\mathbf{c}}, \mathbf{s}'_{\backslash j}, s_j, \mathbf{r}) P(\tilde{\mathbf{c}}, \mathbf{s}'_{\backslash j}, s_j|\mathbf{r})$$

where the notation $\backslash j$ indicates all indices except $j$. The sum over $s_j$ is simply $P(\tilde{\mathbf{c}}, \mathbf{s}'|\mathbf{r})$ and the sum over $j$ yields a factor of $N_{odors}$; that factor exactly cancels the $N_{odors}$ on the first line. Thus, in the limit $dt \to 0$,

$$\sum_{\mathbf{s}} \left[ \prod_j T(s'_j|\tilde{\mathbf{c}}, \mathbf{s}, \mathbf{r}) \right] P(\tilde{\mathbf{c}}, \mathbf{s}|\mathbf{r}) = P(\tilde{\mathbf{c}}, \mathbf{s}'|\mathbf{r}) \,. \tag{2.4}$$

Figure 2: Log probability and probability codes make different predictions about activity of the non-presented odors. The left column is $\log p(s)$ for the variational algorithm; the right column is $p(s)$ for the sampling algorithm – exactly the same as the right column in Fig. 3 of the main text. For the variational algorithm, the activity of the neurons ($L$) codes for log probability (relative to some background to keep firing rates non-negative). For this algorithm, the drop in probability of the non-presented odors from about $e^{-5}$ to $e^{-12}$ corresponds to a large drop in firing rate. For the sampling based algorithm, on the other hand, activity codes for probability, and there is almost no drop in activity.

Figure 3: Same as Figure 3 in the main text, but for a longer time (up to 1 second) and up to 10 presented odors. Increasing the number of components in the mixture reveals the advantage of sampling over the variational approach — on average, sampling makes slightly more correct guesses than the variational algorithm.