[Reviews · NeurIPS 2013]

Submitted by Assigned_Reviewer_4

This study investigates two algorithms for fast inference in generative models of olfaction. Their goal is to compute the most likely, linear mixture of odors comprising an olfactory stimulus. One of the algorithms employs variational inference, while the other is based on a sampling scheme. Simulations demonstrate that both algorithms perform suitably well, and the authors claim that inference is performed rapidly within the first 100 ms, while eliminating false positives (detection of odors not present in a particular stimulus) takes much longer and is difficult when more than two odors are present. As the inference schemes are fundamentally different, it is concluded that in an experiment, it would be possible to decide which of these alternatives is used by the brain.

The manuscript is nicely written and touches an interesting topic. The mathematical section is dense, but technically sound, and it makes realistic assumptions about the problem that the olfactory system has to solve.

The most problematic issue is the claim that inference can be performed within equally short time intervals as in the brain. It is based on the time constants introduced in equation 3.10 which are choosen to match neural time constants. However, on the right hand side of these DEQs there are many nonlinearities (log and exp) - and it is not clear whether a neural system would be able to perform these operations with the required precision in the time given by these time constants. Similar remarks also apply to the second, sampling-based algorithm. The authors state in the discussion that a realistic neural implementation would probably decrease performance, but I think that the problems with timing would be even more severe.

A second issue which is partly unclear concerns the evaluation of the two models against experimental data, in particular if both models will be implemented as spike-based networks. Here, the discussion is a bit vague - what exactly has to be measured in order to distinguish between the sampling and variational inference hypothesis?

In summary, I think the manuscript is technically very interesting, but whether it provides a suitable approach to explain olfaction in the brain remains doubtful.

Minor comments:

Figure 1: the curves are barely distinguishable from the axes - try to use a logarithmic vertical axis, don't stretch the figure over the whole page.

Typo, after Eq 3.11: auxillary variable

Typo, remove full stop in sentence before Eq. 3.14

Typo, before Eq. 3.17: for each odour
Summary: An analysis of two algorithms for fast inference in generative models of olfaction is presented. While sound and interesting some doubts remain about its explanatory potential.

Submitted by Assigned_Reviewer_7

This paper considers two competing hypotheses of how the olfactory system could infer the presence and concentration of odors given the firing rates of olfactory receptor neurons responding to a complex, mixed scene. A simple probabilistic model is defined and two inference algorithms are proposed: one based on variational inference and another based on sampling. Dynamical update rules are derived in order to formulate these algorithms in a neurally plausible manner. The biological feasibility of these two algorithms is then assessed using simulated data. The authors conclude that either algorithm could hypothetically work within natural time constraints, and that the methods would be distinguishable based on observed firing patterns.

This paper focuses on a popular theme at NIPS and within the broader computational neuroscience community, namely whether and how probabilistic reasoning and computation could be performed in neural systems. The authors extend a probabilistic model for olfaction introduced at NIPS 2012 (Beck et. al. [1]) by introducing a spike-and-slab prior over odor concentrations, and introduce two novel inference algorithms. Bayesian inference in these models is nontrivial, so developing neurally-plausible implementations of such algorithms is a challenge. The authors use a variety of interesting approaches to the problem of olfactory inference and develop novel and insightful proposals.

The primary conclusion of this paper is that variational inference and sampling could both hypothetically work, but the authors provide little corroborating biological evidence to suggest that either is actually employed in real systems, or that the implementation-level requirements imposed by these algorithms are reasonable expectations for biological systems. A further assessment of the computational and anatomical constraints imposed by these algorithms is left open for future work.

The inference algorithms proposed here are substantially different from other sampling-based approaches and contribute interesting ideas to the growing literature on probabilistic inference in neural circuits. However, it remains to be seen whether biological or experimental evidence can be found for probabilistic inference in olfactory systems.
Summary: This paper builds on existing work by introducing a more reasonable probabilistic model and two novel inference algorithms for olfaction. Though technically sound and theoretically interesting, the biological constraints required to realize these algorithms in neural systems are substantial.

Submitted by Meta_Reviewer_3

Comments:

This paper describes two candidate mechanisms for implementing
probabilistic inference in the olfactory system, one based on
variational inference and a second based on sampling. The basic
problem is to infer the (marginal probability over) the identity and
concentration of odors present in a sparse mixture from the noisy
responses of a population of olfactory receptor neurons. The ORNs
respond linearly with Poisson noise, and the proposed algorithms
perform inference under a spike-and-slab prior on odors.

The problem is clearly motivated and the work itself is very timely
given the recent surge of interest in mechanisms for performing
Bayesian inference in neural circuits (and in the olfactory system in
particular). The ideas are novel and the performance seems impressive;
I'm surprised by the improvement over template matching. This is
certainly one of the more creative and more interesting papers I
reviewed this year.

The paper seems well suited to NIPS insofar as it brings together
theoretical ideas about algorithms for probabilistic inference and
practical ideas about how to implement them in neural hardware (a
consideration that does not arise in standard machine learning
papers).

Comments:

- Variational algorithm: the intuition for the steps in deriving the
variational approximation (pg 3) are not exceedingly clear, and some
of the assumptions seem like they might be rather severe. In light of
this, it seems surprising that the variational algorithm works so
well, that is, that the sampling-based estimate isn't more accurate.

- Sampling algorithm: some clever ideas here. I like the trick for
changing variables to \tilde c_j and the idea for asynchronous Gibbs
sampling. However, the relationship between s' and \tilde c (the
variables being sampled) and neural activity in the network should
be spelled out more clearly.

- Concern about plausibility: the inference algorithms both work in
real time, but the ORN responses r_i seem to be fixed (discrete)
spike counts, which doesn't seem well matched. Can the same
algorithms be applied with point process ORN responses? Also, it
would help to include some detail about how realistic the
assumptions about connectivity and number of ORNs is (for
simulations described in Section 4).

- It would be nice to have some more explicit predictions about the
form of the neural activity itself (i.e., as opposed to predictions
about the relationship between neural activity and probability).
The prediction of "increased variability" for sampling-based
inference doesn't seem very specific, since one could also implement
a dynamical system for variational inference in which the individual
responses are very noisy. Are there any data currently in existence
that might be compared to model predictions?

Minor comments:

- eq 2.2a: subscripts on alpha and Beta seem slightly puzzling,
especially since they're missing in eq. 2.3. (It's not till we see the
variational approximation section that we see why they need to be
distinguished from alpha_0 and Beta_0).

- eq 2.2b: should say what \pi is

- pg 2, line 098: not clear what it means that alpha_0 and and Beta_0
can model the effects of the baseline firing rate.

- pg 2: clever use of multinomial theorem, but the N_ij need clearer
definition. The "N_ij is shorthand for..." sentence is very
confusing. Indices of summation (j=1 to to r_j?) should be given in eq
3.2.

- I wonder if there's any particular reason to use Langevin sampling
(as opposed to either vanilla Metropolis-Hastings or something
fancier).
Summary: Nice combination of theoretical ideas about algorithms for probabilistic inference and practical ideas about how to implement them in neural hardware. Plausibility could be justified more strongly and details of the implementation in neural circuits could be fleshed out a bit more.
Author Feedback

Author rebuttal: The major concern raised by the reviewers was lack of contact with biology. We fully agree that this is a problem with the paper (and we return to that point shortly). However, we would first like to point out that our main goal was to provide a complete formulation of two competing models: one based on sampling, and the other on probabilistic population codes. There is an ongoing debate about which one the brain uses (Fiser et al., 2010 in TICS; Ma et al., 2006 in Nat Neurosc; Berkes et al., 2011 in Science;), so this is an important and timely issue. As far as we know, this is the first time the two models have been applied to the same problem. Ultimately this should pave the way for experimental tests. In the revised version, we will make it much clearer what our goal was.


That said, biological plausibility and experimental tests are certainly critical components of this work. While we are not far enough along to make detailed physiological predictions (that would probably require a spiking neuron implementation), we will expand on the following issues:

1. Both models predict that there should be neurons that code for the presence or absence of odors, independent of concentration. If such neurons are not found, both can be eliminated, at least in their present form.

2. Spikes mean different things in the two models. For the sampling based model, the probability that an odor is present is proportional to firing rate; for the variational model, the probability is a sigmoidal function of firing rate. One could use behavior – say fraction correct on a two-alternative forced choice task – as a measure of the probability that an odor is present, and plot fraction correct versus firing rate. A linear relationship would be strong evidence for the sampling based model; a sigmoidal relationship would be strong evidence for the variational model (at least as formulated here).

3. We are prepared to propose a mapping onto olfactory bulb and piriform cortex (as in Beck et al. (2012)), but we realise it would be just one of many choices to make, and thus of limited explanatory power. Until experimental data dealing with representations of complex mixtures of odors both in the bulb and the cortex is available, our mapping remains a speculation. Indeed, one of the motivations for this work is to encourage physiologists to present more complex and difficult to parse olfactory scenes.

4. As if to emphasise this point, for the simple olfactory inference problems considered in this work, both algorithms turned out to be very fast, and therefore indistinguishable in terms of the average time-course. In a revised paper, we will include a figure addressing a more difficult problem of olfactory inference for which there is a distinction between these two models.


Although our dream experiments remain to be performed, we can make some contact with recent findings:

We will discuss the work of Miura et al. (2012), who provided evidence for a simple rate code (spike count code) in the piriform cortex that is in agreement with both our models. Similarly, experiments on pattern completion and separation by Chapuis and Wilson (2012) indicate that odor identities are more likely to be represented in the anterior piriform cortex. Additionally, Miura et al. (2012) showed that noise-correlations were quenched after odor presentation, an observation which is naturally consistent with the sampling hypothesis.

We agree that these experiments should be acknowledged in our manuscript, if only to demonstrate the difficulty of the distinction we are trying to make.


The proposed changes require that we relegate some of the derivations to supplementary material. However, this will also allow us to more comprehensively describe all the mathematical derivations.


We appreciate all the specific comments, we will implement them. We address the remaining major comments in the following.

Reviewer 1
We will discuss how the various nonlinearities in our models can be implemented. Certainly, some approximations will need to be used for log/exp mapping in their highly non-linear range. To some extent, non-linear transformations can be performed by dendrites (Poirazi et al., 2003; Ujfalussy and Lengyel, 2011). As to divisive normalization - it has already been observed in the olfactory bulb (Olsen et al., 2010).

Reviewer 2
We will augment the discussion of the constraints and assumptions of our encoding schemes. In particular, we will point out that sampling requires more ”basic” computations than the variational scheme, such as addition and division, but it also requires more sophisticated connectivity - ”links” that influence inputs and outputs of concentration-coding neurons. We will acknowledge existing work on neurally plausible implementation of sampling by Buesing et al. (2011).

Reviewer 3
We agree that using the true posterior as a benchmark for our algorithms would be ideal. Unfortunately, this can be done only if the number of odors is very small. Likewise, it would be interesting to study how the results scale with network size, and we have looked at this for our sampling-based model (unpublished data; we found that it generally scaled well). However, the results depend strongly on how we scale the weights. Because of this, and because of the NIPS page limit, we decided against an investigation of scaling.

Reviewer 4
We thank the reviewer for the positive review and valuable insight.

We thank all reviewers for their detailed feedback; we believe it will lead to substantial improvements in the paper. We were also happy that the reviewers thought our work may have high impact. We hope that this reply will tip the scales to allow us to present this work to the NIPS community.